# ThirdPeak is a flexible tool designed for the robust analysis of two- and three-dimensional tracking data
Thomas Müller [1], Elisabeth Meiser[1,2] & Markus Engstler [1,2] ✉

Biological processes, though often imaged and visualized in two dimensions, inherently occur in at least three-dimensional space. As time-resolved volumetric imaging becomes increasingly accessible, there emerges a necessity for tools that empower non-specialists to process and interpret intricate datasets. We introduce ThirdPeak, an open-source tool tailored for the comprehensive analysis of two- and three-dimensional track data across various scales. Its versatile import and export options ensure seamless integration into established workflows, while the intuitive user interface allows for swift visualization and analysis of the data. When applied to live-cell diffusion data, this software demonstrates the benefits of integrating both 2D and 3D analysis, yielding valuable insights into the understanding of biological processes.

Time course imaging is an invaluable tool for measuring the behavior of larger organisms, single cells, and even single molecules. However, probing these systems, with their significantly varying scales, necessitates specialized imaging setups. Although two-dimensional imaging is typically the standard in biology, it only captures a projection of a three-dimensional process, which can lead to misinterpretations of the underlying phenomena. As advanced imaging hardware and storage space become more accessible, generating three-dimensional time course data has become more feasible. These larger and more complex datasets require innovative tools to aid non-specialists in visualizing and interpreting the acquired biological processes.

Given the significant variation in the scale and organization of imaging data, analysis tools are frequently customized to suit specific imaging modalities. For instance, volumetric image stacks of fluorescent samples are commonly analysed using convenient all-in-one tools such as TrackMate[1] or u-Track 3D[2], dispelling the need to establish an elaborate preprocessing pipeline.

Owing to its high resolution, speed, and well-established localization procedure, a wide array of molecules and systems has been investigated using single-molecule localization microscopy (SMLM). Continuous enhancements in fluorescent dyes[3], camera sensors[4] as well as localization and tracking algorithms[5,6] contribute to the ongoing evolution of this field.

While many processes are readily observable in two dimensions, exploring the third dimension in single-molecule microscopy presents a more intricate challenge, often associated with substantial requirements in optical hardware. An effective technique for determining the axial position of individual particles in SMLM involves modifying the point spread function (PSF), e.g., by adding an astigmatic lens into the emission path, to encode the emitter's position relative to the focal plane[7] without the time-limiting acquisition of volumetric images.

After acquiring the imaging data, the general procedure for analysing dynamic processes usually involves first the identification and localization of the objects of interest. This procedure is often facilitated for fluorescence-based techniques by available software, however, segmentation and identification of brightfield or digital holographic microscopy data are more complex and often require custom analysis strategies. For SMLM, the localization procedure of the emitters can be automated using freely available software such as SMAP[8] or Picasso[9].

Once localization is complete, the newly generated data should consist of coordinates and a timepoint identifier, representing the positions of the objects of interest over time. This data can then be connected into tracks, ultimately revealing the behavior of the tracked objects. As the process of connecting of points into tracks is independent of the data scale, several software tools such as Swift[10], or Tardis[6] can be used interchangeably.

Although existing tools can extract specific information from these tracks, the implementation of interactive data exploration is often lacking. This aspect, however, becomes crucial in the context of biological data, where omnipresent noise can obscure important characteristics. The current tools for interactive two- and three-dimensional data exploration are often constrained by their specialized data structures, limiting their integration into custom workflows. Despite the proposal of a standard for SMLM data[11], its general adoption remains pending. Consequently, there is a notable gap in open and user-friendly applications for exploring and analysing three-dimensional track data from various methodical origins (Supplementary Fig. 1).

¹Department of Cell & Developmental Biology, Biocentre, University of Würzburg, Würzburg, Germany. ²These authors jointly supervised this work: Elisabeth Meiser, Markus Engstler. ✉e-mail: markus.engstler@uni-wuerzburg.de

We therefore introduce ThirdPeak, a software designed to visualize, and analyse three-dimensional track data, with a focus on but not limited to SMLM. The resizable graphical user interface ensures accessibility for a diverse range of scientists and hardware, while the versatile data import and export features facilitate integration of the software into established workflows. We utilize this toolbox to preprocess, visualize and analyse experimental and synthetic diffusion data of the predominant membrane-bound surface protein of *Trypanosoma brucei*. Our findings demonstrate that analysing tracks in three dimensions significantly enhance the understanding and discrimination of biological processes.

## Results and discussion

### ThirdPeak integrates into existing workflows and enables complex analysis for non-expert users

To ensure seamless integration with both existing and future workflows, ThirdPeak operates with established SMLM formats and accommodates custom MATLAB and comma-separated value (CSV) files via an import dialogue. During import, the user is prompted to provide the dimensionality and scale of the data, making the software suitable for a wide range of temporal and spatial scales, as no image data is processed. The minimal parameters required for localizations are the frame number or timestamp and the lateral coordinates for two-dimensional, with additional axial coordinates for three-dimensional analysis. For already connected track data an additional unique track identifier is necessary (Fig. 1a, Supplementary Fig. 2). This flexibility allows ThirdPeak to accommodate diverse datasets, enhancing its utility across various research applications

The preprocessing stage (Fig. 1b, Supplementary Fig. 2) enables batch-processing of localization data obtained from external workflows that analyse available image data, as ThirdPeak does not provide a localization algorithm by itself. Localization data can be filtered based on position, localization precision, and intensity to remove datapoints of insufficient quality, thereby reducing the computational load during further processing. To address the prevalent issue of drift in small-scale data, drift corrections - with and without fiducial data - can be applied. With the installation of Swift[10] on the system, tracking of localizations becomes seamlessly achievable. Alternatively, parameters such as the distribution of localization precision and the diffraction limit are saved and can serve as input parameters for a preferred tracking algorithm. After each preprocessing step, the data can automatically be saved as a CSV file and used with other software if only a subset of the preprocessing features are required by the user. This approach aims to facilitate the integration of new or highly specialized tracking algorithms that might be required by the experiment.

Using track data facilitates the computation of estimated displacement and bleaching probability of the particles. These values subsequently can contribute to refining the tracking process iteratively.

During the validation phase (Fig. 1c), users can interactively navigate through both localizations and tracks. This data can be overlaid onto the original image sequence, which may consist of either single images or image stacks. While other tools might offer more sophisticated validation parameters[2], visual inspection remains a universal approach across all image modalities, facilitating the identification of any necessary adjustments in either the localization or tracking steps. If the tracking data shows the anticipated connections, users can then proceed to the visualization window for a comprehensive data analysis (Fig. 1d).

The central window offers the flexibility to manipulate the three-dimensional data, enabling exploration from various viewpoints. On the right-hand side of the visualization window, users have the option to manually subset the data and save the selections for later analysis (Supplementary Fig. 2). These subsets can either contain a selection of single tracks or whole regions of interest, including only the datapoints within this region or all datapoints of tracks traversing the region.

The analysis supports three data modalities. If only the minimal parameters for the timepoint and coordinates are provided, the software aims to derive well-established standard parameters, such as jump distances and angles, as well as diffusion coefficients. Users with additional data from

their chosen tracking software can input it during the import dialogue, making it available for plotting and exploration within ThirdPeak. As a third option, we integrate the Python-based package DeepSPT[12], a diffusional fingerprinting and classification tool, to study anomalous diffusional behavior in greater detail. We compared the accompanying classification network of DeepSPT with a self-trained network, using artificial data mimicking the temporal and spatial dimensions in our experiments (Supplementary Fig. 3). Both networks performed nearly equal with the same test data, with the self-trained network achieving better results when identifying anomalous diffusion, while the provided network was better in classifying confined motion. With decreasing track length, correct classifications of both networks became less but remained above the threshold of random classification. The additionally calculated parameters can also be of use for regular analysis. Skilled users are encouraged to use this Python interface to implement their own workflows, enhancing the flexibility and functionality of ThirdPeak.

A data filtering feature is available for all properties, enabling users to further refine the data based on their specific interests. This functionality allows for targeted analysis of concealed dynamics within selected groups. These data filter settings can be saved and loaded, thereby enhancing the reproducibility of the data analysis (Supplementary Fig. 2).

The provided and internally calculated data is typically visualized in histograms. Parameters such as track lengths and jump distances allow additional statistical interpretation through boxplots or Gaussian distribution fitting. For example, the distribution of the mean jump distance can help identify distinct diffusing populations if other parameters are unavailable. We have further implemented the jump distance distribution analysis proposed by Menssen et al.[13] to provide an alternative method in determining the diffusional properties of the sample.

The software allows users to flexibly combine multiple loaded datasets for a comprehensive analysis. Although direct comparison of two dataset combinations within the software is not possible, data used to generate a histogram can be immediately exported as a CSV file and imported into a software of choice. This approach usually offers user with greater flexibility in visualization and statistical options.

Additionally, users can produce heatmaps with the calculated data by binning the data in time and space, facilitating the identification of spatially confined behaviors in cellular microdomains. Further, we provide a HDBSCAN functionality allowing users to identify clusters within a selection of parameters of their choosing.

The entirety of the data can be exported as a CSV file or in Excel formats. Moreover, the generated figures can be exported as a scalable vector graphic, suitable for presentations and publications. This streamlined process provides a swift and accessible method for exploring and comprehensively analysing three-dimensional track data, eliminating the necessity for programming expertise.

In summary, we provide a flexible, modular toolbox that allows the exploration of complex three-dimensional datasets, regardless of the acquisition method. With a simple yet powerful graphical user interface, we empower non-expert users to visualize and interpret their data effectively.

### Three-dimensional data analysed with ThirdPeak reveals additional dynamics on the surface of *Trypanosoma brucei*

To demonstrate the software's capabilities, artificial diffusion data of three mixed populations was generated using SMIS[14] (Table 1). The simulations were created as an "ideal" set of images containing no additional noise. The 50 active emitters were designed to have an infinite lifetime and to move with three different diffusion coefficients ($0.16\,\mu m^2 s^{-1}$, $0.56\,\mu m^2 s^{-1}$, and $1.33\,\mu m^2 s^{-1}$) in a limited, cell-shaped environment, presenting a challenging scenario for the tracking algorithm. Additionally, experimental data was acquired from living, immobilized *T. brucei*, using astigmatism to encode the axial position (Supplementary Figs. 4, and 5). In these experiments, fluorophores are usually sparser and show a limited lifetime. Single molecules from the experiment and the simulation were localized using SMAP, followed by preprocessing of the localization data in ThirdPeak. The filtered

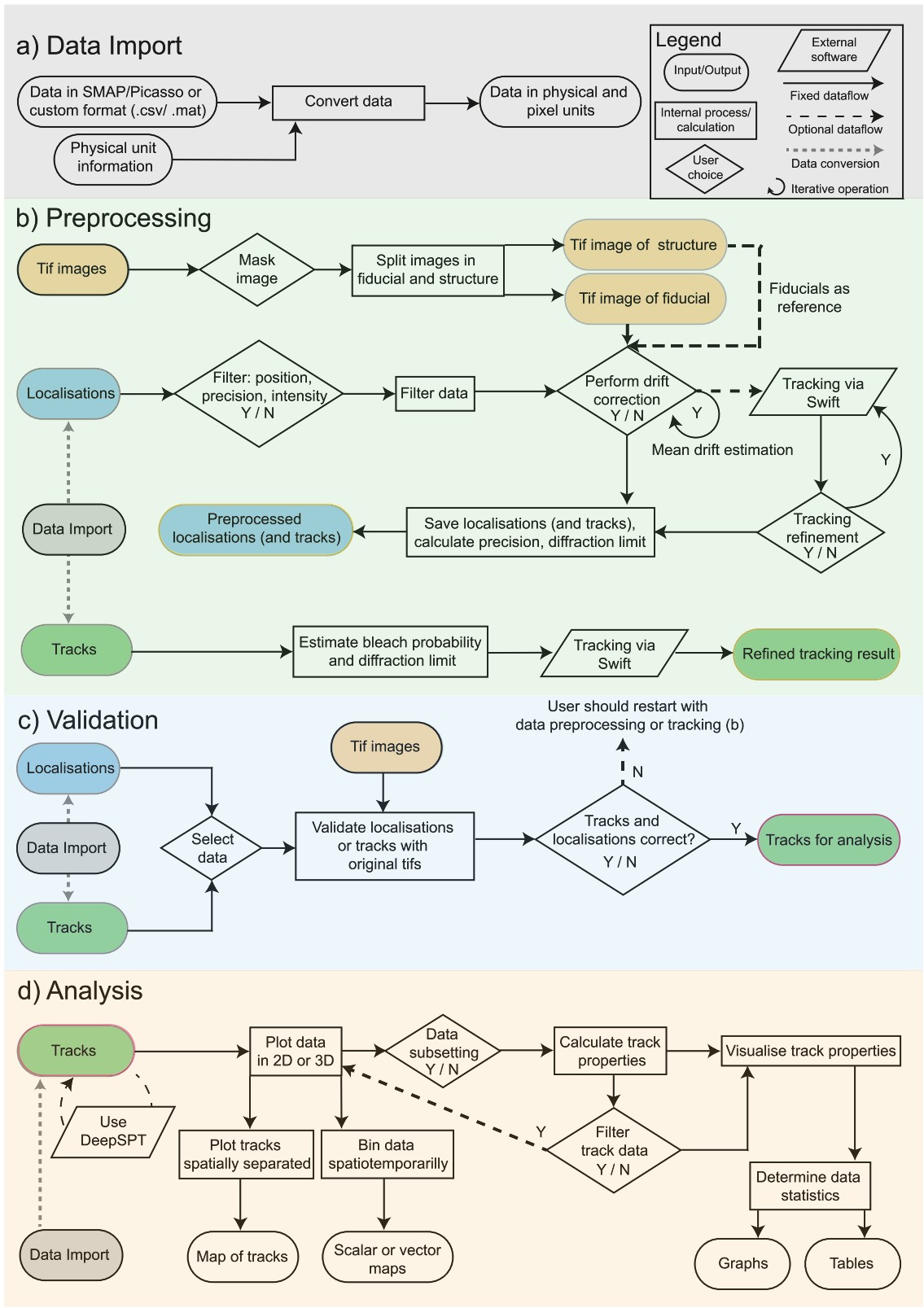

data was automatically tracked using the Swift integration and subsequently visualized and analysed using ThirdPeak again (Fig. 2, Supplementary Fig. 5).

A comparison of track statistics between the simulated and experimental datasets reveals only minor differences in the number of steps tracked and the total track length. The simulations generally produce longer tracks, as no bleaching occurs. Given the small difference in length, it is assumed that the dense tracking environment promoted ambiguous connections, guiding the tracking algorithm to separate tracks rather than create one continuous track. Notably, when examining the net track length (the distance between the first and last localization), the experimental data exhibits lower values compared to the simulation. The experimental data is

**Fig. 1 | Data flow for the data import, preprocessing, validation and analysis in ThirdPeak.** Illustrating the input and output steps (ellipses), the internal processing of the software (rectangles), the choices for the user (diamonds), and the external calculations (parallelograms). The different arrow types describe the predetermined data flow (solid line), optional data flow (dashed line), the data conversion performed using the parameters set in the import window (gray dashed line) and iterative, recursive behavior (circular arrow). Y/N: Yes/No. **a** *Data import:* The user defines the physical units for the pixel size and the frame time, as well as the structure of the data to be imported. With this information, the software can convert the data into the internal format. **b** *Preprocessing:* Either localization or track data might be loaded. For localizations, data can be filtered by their coordinates, localization precision or intensity. Drift correction can be performed using either fiducial markers from the original images, or by a mean drift estimation approach. Preprocessed localizations can be tracked using Swift or saved to be used in a tracking software of choice. Already constructed tracks can be loaded to estimate additional parameters than could help to refine the tracking process. **c** *Validation:* To validate externally generated localizations or tracks, the respective data can be overlaid with the original images, allowing the visual inspection of the quality of the generated data. If the data does not show the expected quality, users are advised to return to the preprocessing step to redo the localization and tracking. **d** *Analysis:* Users can load their track data and decide if they want to generate additional track properties using DeepSPT. The loaded data can be spatially separated into subsets and analysed. Alternatively, track properties can be averaged in spatiotemporal bins to reduce inherit noise in the data. Tracks can be filtered by their properties and several property filters can be connected via logical statements. Finally, data can be visualized in graphs and tables and involved data can be exported in CSV files for further analysis.

## Table 1 | Properties used to simulate 2D and 3D microscopy data

| Property | Value | Property | Value |
| --- | --- | --- | --- |
| Image size XY | 95 | Image size Z | 23 |
| Use drift | 0 | Obj_NA | 1.47 |
| Obj_DOF | 500 | Obj_Immersion | 1.515 |
| Obj_Immersion_Sample | 1.33 | Obj_transmission_eff | 0.2 |
| Obj_mic_transmission | 0.8 | Psf_n_zslices | 32 |
| Psf_astigmatism_x | 2.5, 1.25 | Psf_astigmtaism_y | −2.5, 1.25 |
| Laserprofile | Gaussian | FWHM | 32 |
| Laser_wavelength | 640 | Fluorophore_extcoeff | 150,000 |
| Fluorophore_extcoeff_lambda | 646 | Fluoroph_quantumYield | 0.65 |
| Number of states | 1 | Name of states | Fluorgenic |
| Number of fluorophores | 50 | FlurophorMotionD | 0.16, 0.562, 1.322 |
| EMCCDGain | 150 | EMCCD_QE | 0.94 |
| EMCCD_offset | 100 | EMCCD_e2ADU | 10.7 |
| EMCCD_readoutNoise | 74 | | |

typically noisy, and stringent filtering conditions for the localizations may result in the loss of some intermediate localizations, leading to shorter tracks. An experimental explanation for this could be the potential presence of either confinement or fast bleaching processes, which are not present in the "ideal" simulation (Supplementary Table 1). The diffusion coefficients through fitting to the cumulative squared jump distance distribution and are in good agreement with previously determined 2D diffusion data[15] (Supplementary Fig. 5, Supplementary Table 1). The focus lies on the first diffusion coefficient rather than the effective diffusion coefficient, as mainly confined particles in the experiments can account for more than 50% of the available datapoints. Plotting the diffusion coefficients of the simulated data in a histogram allows to identify the initial parameter (Supplementary Fig. 7). Binning in velocity maps helps to reduce noise, which is omnipresent in experimental biological data, by spatial averaging. In the ideal simulated data, an unusually slow region is visible in the center (Fig. 2). This is because we designed the simulations to include slow-moving particles that follow the principle of Brownian motion, randomly exploring the space remaining in a region for a longer period compared to the faster-moving particles. Since these slow-moving particles neither bleach nor move away, they significantly contribute to the local diffusion coefficient, resulting in the slow region value. When the data is split into several temporal bins using ThirdPeak, a less averaged overview of the local diffusion coefficients becomes visible (Supplementary Fig. 7), providing more detailed insights into the variations over time. In the experimental data, an increased velocity is frequently observed along the groove of the attached flagella. The mean jump distance analysis can reveal either two or three diffusing populations, depending on the dimensionality of the data.

To the best of our knowledge, ThirdPeak is the first software to seamlessly integrate interchangeable 2D and 3D track analysis in a highly modular yet accessible way. As the toolbox can operate independently of the imaging and tracking data, it also suitable for data modalities not typically covered by all-in-one packages, such as digital holographic microscopy[16] or two-camera setups used to track large biological objects[17], which require their own localization and potentially also tracking approaches. We anticipate that ThirdPeak will prove beneficial for users of different backgrounds. Our commitment includes ongoing efforts to improve the software by incorporating additional features in future updates.

## Methods
The software was developed using MATLAB 2023a using Windows 10.

### Simulations
The simulation of the single molecule data was done using SMIS2.1, running in MATLAB 2022 in Windows 10. Imaging parameters were chosen to represent the local single-molecule microscope.

For the image generation, we used the following properties for the simulation. The difference between 2D and 3D simulations are the "simul_3D" value, which was 1 for 3D simulations and the diffusion environment used for the particles.

To generate a simulated diffusion environment in both 2D and 3D, an initial widefield image of *T. brucei* was utilized. The image underwent processing in Fiji, including Gaussian blurring with a width of 0.5, followed by edge detection and binarization using the Otsu method. The "Fill Holes" command was then applied to obtain a closed shape of the parasite, where the value 1 represented the parasite, and 0 represented the outer area. This

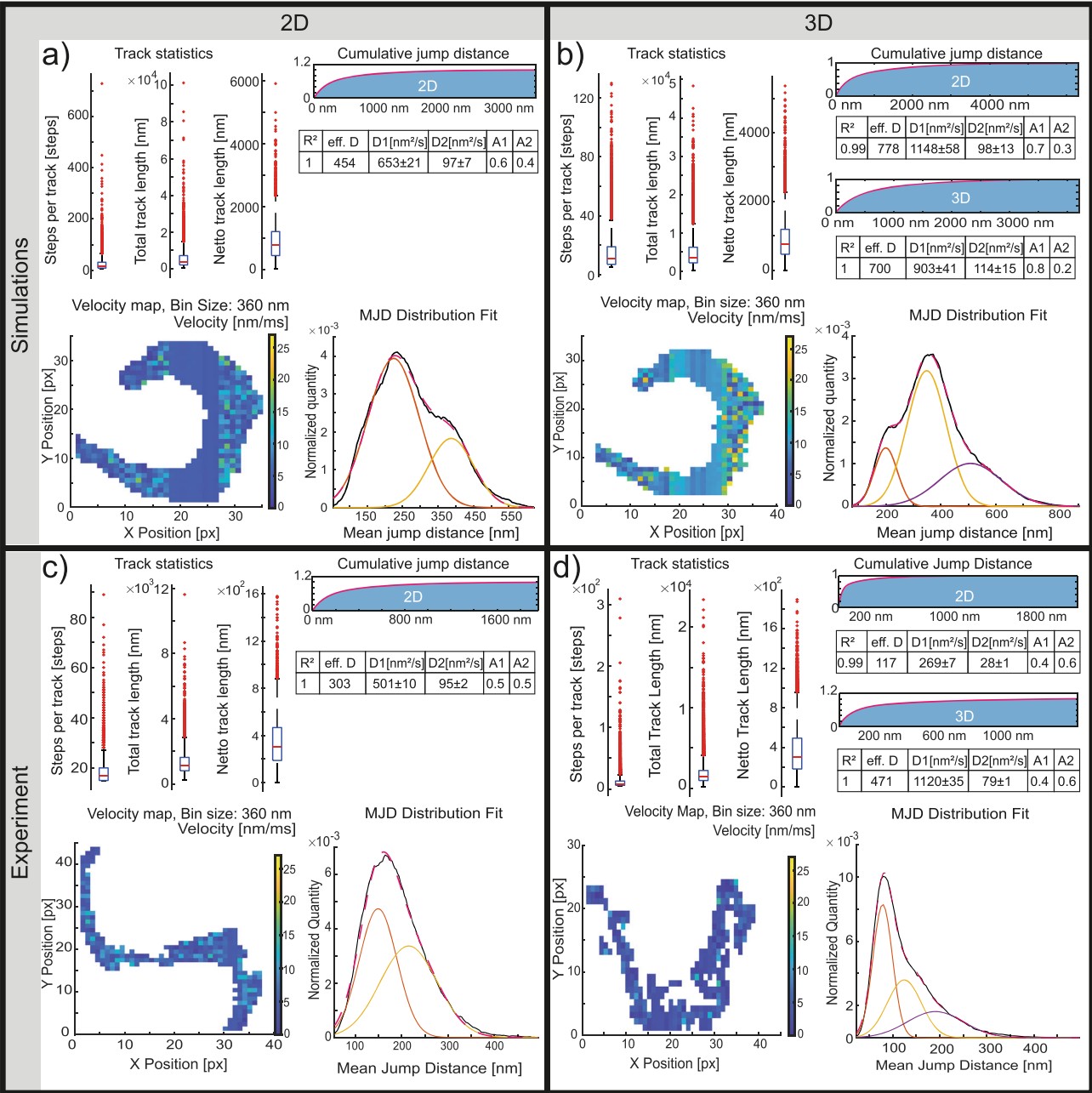

**Fig. 2 | Simulated and experimental diffusion data of the surface coat of *Trypanosoma brucei*.** Overall track statistics showing the median (red line), the 25th and 75th quantile, whisker denoting the 1.5× interquartile range and outliers (red crosses), a fit to the cumulative jump distance distribution to determine the diffusion coefficients and their standard error of the mean, a velocity map and the mean jump distance distribution are shown. **a** In the two-dimensional simulations, the median step length of about 12 steps, which is larger compared to the experimental data (6). The cumulative jump distance fit aligned well to data and identified two diffusion coefficients. In the velocity heatmap, a prominent, slow area is visible in the center due, to a predominant, slow-diffusing fraction that masks faster processes. Using a Gaussian distribution model on the mean jump distance distribution helped identify two diffusing populations. **b** For simulated three-dimensional data the track statistics are similar to the two-dimensional simulations. The fit of the cumulative jump distance also aligned well and identified two diffusing populations. The heatmap shows a slow region in the center, again due to the slow diffusing population strongly

contributing to the local diffusion coefficient. The mean jump distance distribution fit allowed us to identify three of the diffusing populations present in the simulation. **c** In the two-dimensional experimental data, the track length is shorter compared to the simulations. As experimental data can be strongly impacted by confined particles, the effective diffusion coefficient is usually slower; however, the first diffusion coefficient is within the lower range of the values previously determined on the surface of trypanosomes. With less data used for the heatmap, it is less averaged and shows a more heterogenous distribution of velocities. The mean jump distance distribution again allowed the identification of two diffusing populations. **d** In the three-dimensional experiments, the overall track length does not differ considerably from the two-dimensional setup. The first diffusion coefficient is twice the size of the two-dimensional data, but still within the upper range of previous experimental data. The overall velocity appears to be dominated by the slow-diffusing population, which is also evident in the mean jump distance distribution.

binarized image was saved and imported into MATLAB to create a 3D diffusion environment.

In MATLAB, the "erode" command was employed to systematically reduce the size of the shape at its edges. This erosion process was repeated

four times until the remaining shape resembled the center of the original parasite image. Each iteration of erosion was saved and used to construct additional layers of the 3D object. The original binarized image was placed in the center, with four layers added on top and four layers added below to

mimic a three-dimensional trypanosome. Voxel values of 1, representing the parasite, in contact with the outside (value of 0) were assigned a value of 2 to denote them as membrane voxels. The resulting model was saved in a dataframe format suitable for use in the SMIS software.

## Cell culture and cell line generation

*T. brucei* strain Lister 427 were cultivated at 37 °C, 5% $CO_2$ in HMI-9 (*Hirumi's Modified* Iscove's medium-9) containing 10% heat-inactivated fetal bovine serum.

## Sample preparation and imaging

Thickness-corrected coverslips were placed in coverslip racks made from Teflon and cleaned 2 times using 2% Hellmanex-II solution by incubating them for 10 min in an ultrasonic water bath (Elmasonic P, Elmasonic) using 100% power, a frequency of 37 kHz at room temperature. Coverslips were then washed using $ddH_2O$ and stored in $ddH_2O$ until use. On the day of the experiment, two coverslips per sample were dried using an air stream and placed in a clean petri dish. For one experimental day, $1 \times 10^7$ cells were harvested by centrifugation at $1500 \times g$ for 10 min at room tempreature. The supernatant was removed, and the cells resuspended in 1 ml of vPBS (PBS supplemented with 10 mM glucose and 46 mM sucrose, pH 7.6). Cells were washed 3 times with 1 ml of vPBS at $800 \times g$ for 2 min, followed by resuspension in 100 µl vPBS, containing 1 nM Atto643-NHS (ATTO-TEC), and incubation for 5 min on ice. The dye labels the variant surface glycoprotein coat of the parasites. Following incubation with the dye, the cells were washed 3 times with 1 ml of vPBS at $800 \times g$ for 5 min at 4 °C. Finally, the cells were resuspended in 20 µl vPBS and stored on ice. As trypanosomes are highly mobile parasites, they need to be immobilized for imaging. For this, a mixture of 8-arm PEG-VS (2.5 µl of 50 mM solution, Sigma-Aldrich) was used together with hyaluronic-acid functionalized with an SH group (8–10 kDa, DS 40%, 3 µl of 25% solution, AG Groll[15]). Additionally, 1 µl of 1:100 diluted latex beads (BD Biosciences) of 6 µm diameter were used as a spacer to guarantee equal spacing of the coverslips. As a buffer 2 µl of vPBS and 2 µl of cell suspension were mixed and added onto a coverslip. A second coverslip was placed on top, and a weight of 90 g was applied to the sample. The coverslip sandwich was then centrifuged for 1 min at $1500 \times g$ to collect the trypanosomes at the lower coverslip.

Imaging was done at 37 °C on an inverted widefield microscope (Leica DMI6000B) with an Olympus 100 × oil immersion objective (HCX PL APO 100 × 1.47 OIL CORR TIRF) and a dichroic filter (zt405/514/633rpc, 670, Chroma). Fluorescence was excited using a laser beam (Hübner Photonics) at 640 nm with a power of 3 kW/cm². Pulsed illumination was controlled using an acusto-optical tunable filter AOTF to reach illumination times of 9 ms or 36 ms at frame rates of 100 Hz or 25 Hz for 2D and 3D, respectively. Either 2500 or 5000 frames were acquired. Image acquisition was performed using an Andor iXon697 with a gain of 150, running in cropped mode with a ROI of 120 × 120 pixel. The acquisition process was controlled using MicroManager 2.0.

For calibration of the 3D astigmatic approach, TetraSpeck™ (ThermoFisher Scientifc) beads were immobilized using the PEG-HASH hydrogel and z-stacks with 10 nm step size were acquired.

## Image processing

Calibration image stacks of fluorescent beads were processed in SMAP to determine the astigmatism using a spline fit. The resulting calibration file was used for z-position determination in the experimental image data. For this, the batch mode of SMAP was used. The localizations were then loaded into ThirdPeak, and a manual mask was applied to only include trypanosome cells. Next, localizations were filtered by their precision ($\sigma_{XY} \leq 100$ nm, $\sigma_Z \leq 200$ nm) and corrected for drift using a mean-shift approach. The resulting localizations were saved as CSV files for tracking analysis by swift. The data were loaded into ThirdPeak to refine the tracking by adjusting the expected displacement and bleach values accordingly. Tracking was optimized by iterating the tracking process until the expected displacement

between 2 loops differed less than 5%. These optimized tracks were then loaded into ThirdPeak for further analysis.

During optimization, the tracking process converged the expected displacement value for the two-dimensional simulation to 217 nm, for the three-dimensional simulations to 290 nm and for the experimental data to 170 nm due to the immobile particles present in the experiment.

## Tracking using swift

For the tracking of the localization data, we used swift (Endesfelder et al., manuscript in preparation, beta-testing repository http://bit.ly/swifttracking), a tracking algorithm relying on Markovian statistics and initial starting parameters. It operates in three stages, the first main tracking stage in which a global optimal linking solution should be found, followed by the pruning stage. During this stage, less likely connections are removed in dense localization regions. In the last postprocessing stage, potential errors that occurred in the main tracking stage can be corrected (swift 0.43 manual).

## Statistics and reproducibility

The experimental data of living, immobilized trypanosomes consist of 40 and 35 cells, respectively for the 2D and 3D case. These cells were acquired from three technical replicates on the same day. For the 2D experiments, 8696 tracks were used for analysis, in the 3D case, 15,020 tracks were analyzed. For the simulations, one file for each condition (2D and 3D) was generated and 4025 tracks and 9590 tracks were analyzed in the 2D and 3D case, respectively.

## Functions in the software

**Heatmap generation.** As biological data is inherently noisy, generating heatmap by binning single-molecule localizations and tracks into larger areas can reveal processes that might just be hidden inside the noise itself. For this, the data range of the tracks is used, and the *x*, *y*, *z* coordinates are placed into respective bins, depending on the user-defined size. Additionally, the user can choose to change the temporal binning by increasing the number of time windows, thereby splitting the data also in the temporal space. This feature detects the directionality of the tracks, the velocity of particles and the number of localizations in each bin.

**Jump distance.** The jump distance describes the Euclidean distance of a particle between two consecutive timepoints in 1D, 2D or 3D, depending on the choice of the user. For one dimension:

$$d_n = x_t - x_{t+1}$$

where *d* describes the distance between two time points in one of the dimensions, xt the positional argument of one dimension at timepoint *t* and xt+1 the positional argument of one dimension at timepoint $t + 1$.

For two dimensions:

$$d_{2D} = \sqrt{(x_t - x_{t+1})^2 + (y_t - y_{t+1})^2}$$

For three dimensions:

$$d_{3D} = \sqrt{(x_t - x_{t+1})^2 + (y_t - y_{t+1})^2 + (z_t - z_{t+1})^2}$$

**Mean jump distance.** The mean jump distance is calculated per track from the jump distances calculated above by:

$$\bar{d} = \frac{1}{n} \sum_{i=1} d_i$$

**Net track distance.** The net track distance describes the distance between the first and last localization of a particle:

$$d_{net} = x_1 - x_n$$

**Total track distance.** The total track distance describes the complete length of a trajectory:

$$d_{tot} = \sum_i d_i$$

**Basic confinement ratio.** The confinement ratio ($R_{conf}$) divides the total track length by the net distance. If $R_{conf} < 1$, the track is not confined. If $R_{conf} > 1$, the track is confined.

**Diffusion coefficient estimation by jump distance distribution.** Estimating the diffusion coefficient from the 1D jump distance distribution is a robust way of calculating this property under the assumption of Brownian motion, especially when having short tracks, as a fit to the mean-squared displacement becomes less reliable. To retrieve the diffusion coefficient, fitting a normal distribution onto the jump distance distribution of one dimension can be performed. The standard deviation from this fit can then be retrieved and by the Einstein–Smoluchowski equation the diffusion coefficient can be calculated by:

$$D = \frac{\sigma^2}{\Delta t}$$

where $D$ is the diffusion coefficient, $\sigma$ is the standard deviation and $\Delta t$ the respective timeframe. This however only determines the overall diffusion coefficient and can not distinguish between different populations.

**Diffusion coefficient estimation by cumulative jump distance distribution.** To distinguish between multiple populations, an analysis using the cumulative jump distance distribution can be performed by fitting the integrated distribution in accordance with Weimann, Klenerman[18] for 2D:

$$P(r^2, \Delta t) = \int_0^{r^2} p(r^2) dr^2 = 1 - e^{-\frac{r^2}{4D\Delta t}}$$

for one species, and if multiple species are present, the sum is used:

$$P(r^2, \Delta t) dr^2 = \sum_{J=1}^{m} \frac{f_j}{4D_j\Delta t} e^{-\frac{r^2}{4D_j\Delta t}}$$

For 3D data, the following applies:

$$P(r^3, \Delta t) = \int_0^{r^3} p(r^3) dr^3 = 1 - e^{-\frac{r3}{6D\Delta t}}$$

**Mean jump distance population distribution.** The mean jump distance population distribution is determined by smoothing the histogram of the mean jump distances with a variable kernel filter and determining the maxima in the smoothed data. The maxima are then used in a minimal least squared approach to fit the data with Gaussian populations.

**Diffusional confinement ratio.** To determine if a particle is confined, an extended ratio in comparison to the basic confinement ratio is possible. The diffusional confinement ratio of a particle is determined by fitting the mean squared displacement with a confined diffusion model:

$$MSD = R^2 * \left(1 - e^{-4D\frac{1}{R^2}}\right) + offset$$

, where $R$ is the radius of confinement and $D$ the local diffusion coefficient.

**Volume calculations.** Volume calculations are performed on the total volume the tracks confine. Two options are available, the convex hull, that forms a convex bounding box around the track data, while the alphaShape approach will fit a tight bounding box around the track data.

**Jump angle.** The angle $\delta$ between two vectors, or essentially three points of a single track is calculated as:

$$\delta = \tan^{-1} ||(axb) - (a \cdot b)||$$

with $a$ and $b$ being the two vectors, respectively.

**Additional code packages.** We further include major features of the following repositories, if necessary, extending their features for 3D analysis and handling (sparse) track data.
- MSDAnalyzer:n https://github.com/tinevez/msdanalyzer/tree/master
- TIFFStack: https://de.mathworks.com/matlabcentral/fileexchange/32025-dylanmuir-tiffstack
- Mean-Shift-Drift-Correction: https://github.com/frankfazekas/Mean-Shift-Drift-Correction
- Peak finding and measurement: https://de.mathworks.com/matlabcentral/fileexchange/11755-peak-finding-and-measurement-2019?s_tid=srchtitle
- TrackIt: https://gitlab.com/GebhardtLab/TrackIt
- Jump Distance Analysis: https://github.com/rmenssen/JDD_Code
- DeepSPT: https://pubmed.ncbi.nlm.nih.gov/38352328/
- HDBScan: https://github.com/Jorsorokin/HDBSCAN

## Reporting summary
Further information on research design is available in the Nature Portfolio Reporting Summary linked to this article.

## Data availability
Data related to this article are available online at zenodo: https://doi.org/10.5281/zenodo.13739340.

## Code availability
The source code as well as compiled versions as a MATLAB app and standalone executable for Windows are available on Github and zenodo: https://github.com/unithmueller/ThirdPeak_2_0/tree/2.0 Version 2.0 was used and is now updated with small bugfixes.

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

## Acknowledgements
We would like to thank Pierre Parutto (Cambridge, UK) and Daniele Bourgeois (Grenoble) for sharing their work and helpful discussion. We further would like to thank Philip Kollmannsberger (Düsseldorf) and Sabine Fischer (Würzburg) for discussion and advice.

## Author contributions
Thomas Müller conceptualized and implemented the software and wrote the initial manuscript. Elisabeth Meiser worked in conceptualizing and proofing/debugging of the software. Markus Engstler supervised the project and provided critical feedback to all version of the manuscript. All authors discussed the project regularly and were involved in the review process.

## Funding

## Competing interests
The authors declare no competing interests.
