## [Transparent Peer Review file · Communications Biology]

ThirdPeak is a flexible tool designed for the robust analysis of two- and three-dimensional tracking data

Corresponding Author: Professor Markus Engstler

Version 0:

Reviewer comments:

Reviewer #1

(Remarks to the Author)

Brief summary of the manuscript

The article introduces ThirdPeak, an open-source tool designed for user-friendly exploration and analysis of 2D/3D single molecule data processed by various localization and tracking software. ThirdPeak aims to bridge a gap particularly in the 3D single molecule localization and tracking field, by offering a GUI-based visualization and analysis platform, thus helping with post-processing and enhancing accessibility to these technologies. Of note, the software is designed for handling 3D data where the z-position is encoded in the shape of the PSF, as for example as obtained via astigmatism, and not 3D stacks. The software allows users to load data processed by algorithms like SMAP, PICASSO, and SWIFT, with the flexibility to work with other data sources after formatting. Note that SWIFT, the primary tracking algorithm used in this work, is cited as “manuscript under preparation”.

ThirdPeak has 3 main functionalities: 1) pre-processing options including drift correction and filtering, 2) visual validation for detection and tracking accuracy assessment, and 3) analysis for generating spatial heatmaps and track statistics. The platform was tested on simulated and experimental data and the track statistics generated by the platform helped identify differences between the two. The simulated data showed overwhelmingly free diffusion, higher diffusion coefficients and longer total track length in both 2D and 3D. They explain this could be due to presence of diffusion barriers in the experimental data or effects of certain experimental settings.

In summary, ThirdPeak offers a GUI-based platform for visualizing and analyzing 2D and 3D tracking data, where the z position for 3D data is obtained via astigmatism. It compiles various pre-existing tracking and diffusion analysis methods and presents them in a convenient drop-down menu format. Moreover, users have the option to conduct analysis on individual tracks or regions of interest directly through the GUI interface. Unfortunately, however, there is some lack of clarity with respect to the scope, as well as the limitations, of this platform. This platform also does not generate any novel statistics or advance on any pre-existing track analysis methods.

Overall impression of the work

Most of the focus of ThirdPeak is on two functions: Validation and Analysis. Its main asset is its ability to handle 3D analysis, which is not common in similar tools. Yet, it must be noted that the 3D data handled by ThirdPeak are the type where the z-position is encoded by the PSF shape. While there are many experimental setups now that treat the z-position in this way, and so ThirdPeak would have good range of applicability, this point is not stated explicitly in the manuscript. This must be stated explicitly, and the authors must clarify whether ThirdPeak can or cannot help with the visualization of 3D stacks, as some other software packages do (e.g., u-track3D by Roudot et al. [cited in the manuscript]).

The Validation functionality allows users to visualize localizations and tracks obtained from other software and check the accuracy of these methods. This feature is not exclusive to ThirdPeak, and is available in various software packages such as SMAP (1), U-track (2D/3D)(2, 3), Spot-On(4) and many others (5, 6). In addition, it is worth noting that the assessments via ThirdPeak are predominantly visual, and hence subjective. In contrast, for example, u-track 3D(3) offers the option to calculate a trackability score for assessing tracking performance, particularly beneficial for 3D tracking evaluations. The authors must comment on this. Where ThirdPeak stands out is in its ability to work with localizations and tracking data from external sources.

ThirdPeak's usefulness lies in its GUI-based analysis platform, simplifying the generation of track statistics and heatmaps, particularly for 3D data. ThirdPeak offers users an opportunity to generate track statistics based on the selected ROI or

single tracks. This ability to select single tracks or regions of interest through a GUI for post processing is an exciting advancement.

A broader concern regarding this work lies with the description of the different features offered by the platform. The text and examples provided lack clarity regarding the extent and limitations of the platform's capabilities. A major concern is that the tracking software used by the authors, SWIFT, is not yet available and is indicated as being in preparation. This makes it difficult to address these concerns effectively. Additional clarification on these specific issues is detailed in the comments section below.

Specific comments, with recommendations for addressing each comment:

Major

1. The authors need to define the minimum amount of information needed by the platform to generate track statistics and create figures. It is not clear from the current text to what extent ThirdPeak depends on external software (like SWIFT) for some basic calculations and plotting their figures. This is especially important for the custom input option (as seen in GUI). Are some inputs optional? Here are some examples highlighting the source of this confusion.

a. Jump distance: Line 61 of the main text states that the platform can calculate the jump distance. Indeed, Jump Distance calculation is included under the "Functions in the software" section. However, GUI seems to also have a "jump distance" input.

b. Diffusion coefficient: Is it primarily obtained through their own calculations (lines 359-379) or by SWIFT (line 227)?

c. Diffusion classification: It appears that the appearance of distinct diffusing populations is primarily performed by the SWIFT (line 216) algorithm, but their own calculations towards classification based on jump distance (line 62-62, 93-94) are also mentioned.

2. Further, related to the scope of the platform, does the 3D visualization and analysis feature work with 3D stacks generated by taking z sections?

3. Authors need to better describe SWIFT, as it is only "manuscript under preparation", according to the authors' citations.

4. While visual comparison between dataset subsets is helpful, it is important to also have access to the actual data used in the visualization, in order to run e.g. statistical tests. This is particularly relevant in the context of running analysis on subsets of the data (ROIs or single tracks). It is not clear at present if ThirdPeak currently offers access to the dataset subsets used for visualization.

5. The understanding of the scope of this work will be greatly improved by a more in-depth explanation of how ROI track statistics are calculated. E.g., in case the ROI happens to include only a snippet of the track/tracks, does the track statistic reflect that of the entire track or only the snippet?

6. The diffusion coefficient and the net/total track length of simulated data appear to be larger than that of experimental data (supplementary table 1). But authors claim that the difference is not large (line 227 and 83-84 respectively). How did the authors come to this conclusion?

7. Figures:

a. Figure 1: Clearly marking out the steps that happen outside of this platform or require input from external sources will aid in understanding the platform better. Fig1C gives the impression that after looking at the tracking/localization overlays one can refine the tracking or localizations within this platform. But this is not the case, after trying out the software. Thus, in its current form, the description of ThirdPeak might be misleading.

b. Also in Figure 1: What do the dashed gray lines in this figure mean? Finally, it is not clear why "localizations" and "tracks" branch out separately out of the "Data Import" box. Isn't localization the first step of tracking? In addition, the "Track refinement" step after "correct drift" seems out of place, as there was no tracking step before it, in order to get tracks to refine.

c. Supplementary figure 3: The figure showing spline fit of calibration bead stack is confusing, why is there an increase in intensity as the no of frames increases? What are frames here, are they the 10nm z steps described in materials and methods? An appropriate legend and further explanation would be helpful.

8. Simulation:

a. Please provide some details about the simulation parameters used for this work.

b. In their reporting of the simulated data results, especially on lines 90-92, the authors present the results of analyzing the simulated data as if they were a "discovery". But these are simulated data that the authors generated, and they must know the answer. The authors should comment on how well their analysis method was able to reproduce the diffusion coefficients used to generate the simulated populations.

c. In lines 86-88, the authors speculate that bleaching or confinement in the experimental data could underlie the observed differences between experimental and simulated data. But, could tracking errors have also contributed to the differences, as experimental data is noisy. Was any noise added to the simulation to recreate noise seen in experimental data sets?

d. The area of very low velocity for the simulated data needs further investigation and explanation. Again, these are simulated data where the authors have full control over the simulation.

Minor comments

1. Figures:

- a. Please provide panel identifiers, they are missing from most of the figures and please provide appropriate (more detailed) legends.
- b. Normalizing the histograms and providing the same range for the color bar for the heatmaps would be very helpful in figures where multiple datasets are being shown and compared.
- c. Figure 2: Diffusion coefficient units are missing. Also, in this figure, the bottom-left subpanel of each panel is called "velocity map", while what appears to be the same figure is called "drift map" in Fig. S4.

2. Please note that the following paper is now published- Roudot, P. et al. u-track 3D: measuring and interrogating dense particle dynamics in three dimensions. bioRxiv 2020.11.30.404814 (2022) doi:10.1101/2020.11.30.404814. Please cite the published version. Also, it will be helpful for the text clarity if this software is referred to as u-track3D throughout the paper, to distinguish it from the original u-track software, which did not provide visualization for 3D images. Please also discuss in more detail the differences between ThirdPeak and u-track3D.

3. The ability to use this platform to compare between multiple datasets in a single figure or plot where applicable would be desirable, as often people are trying to compare these statistics between different conditions. Such a feature would increase the adoption and reach of ThirdPeak. It is not clear from Supplementary figure 5, if the option of being able to choose multiple files for analysis is for batch processing or can one compare between multiple data sets.

4. Please provide some videos (raw data, detections, tracking) of the experiments performed on T.Brucei. It will help readers assess the quality of data, get a sense of how dense the localizations are, especially because the dye is labelling all surface glycoproteins.

5. What were the localization and tracking parameters for the simulation and experimental data sets? Were they the same?

6. If the diffusion coefficients are calculated only in 2D (line 228), then where is the 3D diffusion coefficient calculation used (line 378-379)?

7. The title of the section: Confinement Ratio vs mean jump distance does not match with the description (line 385-389). There is also another reference to the confinement ratio in line 356.

8. On trying the software, the manual was found to contain very little information regarding the use of the canned data. The manual can be made more user friendly by giving examples for every step using the canned data.

-----References-----

1. J. Ries, SMAP: a modular super-resolution microscopy analysis platform for SMLM data. Nat Methods 17, 870-872 (2020).
2. K. Jaqaman et al., Robust single-particle tracking in live-cell time-lapse sequences. Nat Methods 5, 695-702 (2008).
3. P. Roudot et al., u-track3D: Measuring, navigating, and validating dense particle trajectories in three dimensions. Cell Rep Methods 3, 100655 (2023).
4. A. S. Hansen et al., Robust model-based analysis of single-particle tracking experiments with Spot-On. Elife 7, (2018).
5. S. C. Stein, J. Thiart, TrackNTrace: A simple and extendable open-source framework for developing single-molecule localization and tracking algorithms. Sci Rep 6, 37947 (2016).
6. J. Y. Tinevez et al., TrackMate: An open and extensible platform for single-particle tracking. Methods 115, 80-90 (2017).

Reviewer #2

(Remarks to the Author)

Muller and propose a new software suite for the manipulation of 2D and 3D trajectories measured on fluorescence microscopy sequences. The exploration and interpretation of such trajectories remain a complex problem for the community. As such, this topic is of great interest for the community. The paper broadly compares the capabilities of ThirdPeak against the competition, however it is not clear how each part of the pipeline differs from the other tools that are cited. The feature reporting is also sometimes misleading, since both Kuhn and Roudot's paper propose automated approach for validation while Muller's is manual. As such, it appears that the software does not provide conceptual advances but a new package to load, track, visualize and analyze the data. This is in itself can be very valuable, especially considering the dearth of trajectory analysis tools. However, it is impossible to evaluate the value of the new software without actually trying it. To that end, I would need the authors to provide a minimal example of input to test the software. I guess I could try trackIt with my own data to test it, but I respectfully believe that this is out of the scope of the reviewer duty.

Minor comments:

The motivation of the work is in the introduction is unconvincing at time:

- Why taking about PSF engineering in particular ?

- TrackMate and others do not need SMAP or Picasso to produce tracks, they can do so readily on volumetric sequence.

Reviewer #3

(Remarks to the Author)

Version 1:

Reviewer comments:

Reviewer #1

(Remarks to the Author)

I thank the authors for the substantial effort they put into revising the manuscript and clarifying what their software does (and does not do). They have addressed my previous comments, and I have no further critiques.

Reviewer #2

(Remarks to the Author)

I appreciate for the authors revision effort. One remaining important for me is to make sure the software is usable. I currently get stuck at the step highlighted in the file attached. I am happy to try again or discuss debugging.

Reviewers comment	Authors response	Com. Ref
Reviewer #1 (Remarks to the Author):	Thank you for the thorough review of our manuscript and software. Your insightful remarks have helped to enhance the overall quality of our work.	1
The article introduces ThirdPeak, an open-source tool designed for user-friendly exploration and analysis of 2D/3D single molecule data processed by various localization and tracking software. ThirdPeak aims to bridge a gap particularly in the 3D single molecule localization and tracking field, by offering a GUI-based visualization and analysis platform, thus helping with post-processing and enhancing accessibility to these technologies. Of note, the software is designed for handling 3D data where the z-position is encoded in the shape of the PSF, as for example as obtained via astigmatism, and not 3D stacks.	There seems to be a misunderstanding. While we demonstrate the application of ThirdPeak on single molecule data, the underlying concepts of localizations and tracks can be extended across various scales. We designed a versatile data import feature to accommodate user-defined dimensional ranges. Astigmatism is just used as one example. ThirdPeak itself does not include a function for localizing single particles, cells, or organisms, but it can process localization data if X, Y, Z, and T coordinates are provided. We clarify this in the Results in line 80-81. It is correct however that explicit support of volumetric timeseries data was not supported. In the revised, updated version, we added support for volumetric time series data during the validation and visualization steps. When loading a single multilayer TIFF, it is interpreted as a 2D time series. When loading multiple multilayer TIFFs, each TIFF is interpreted as a Z-stack, and the collection of files is interpreted as a time series. We added text in lines 96-97, stating the added functionality. This is also explained in the accompanying manual.	2
The software allows users to load data processed by algorithms like SMAP, PICASSO, and SWIFT, with the flexibility to work with other data sources after formatting. Note that SWIFT, the primary tracking algorithm used in this work, is cited as “manuscript under preparation”.	We chose to use SWIFT for our analysis pipeline because of its speed and ease of use. However, other users may prefer to use their existing tracking pipelines or develop new ones, e.g. with the tools cited in this manuscript. We have no affiliation with the Endesfelder group, so we are unaware of when the final version of SWIFT will be available. It appears that	3

	TARDIS is a newer project from this group. Swift was however already cited in other articles (see. Citation) We clarify this in the manuscript by changing the following lines: 49-51.	
ThirdPeak has 3 main functionalities: 1) pre-processing options including drift correction and filtering, 2) visual validation for detection and tracking accuracy assessment, and 3) analysis for generating spatial heatmaps and track statistics. The platform was tested on simulated and experimental data and the track statistics generated by the platform helped identify differences between the two. The simulated data showed overwhelmingly free diffusion, higher diffusion coefficients and longer total track length in both 2D and 3D. They explain this could be due to presence of diffusion barriers in the experimental data or effects of certain experimental settings.	The functions mentioned in this statement are all present in the current version of the software. We further include track-statistics estimation during pre-processing. These statistics can be used to automate tracking with SWIFT within the software or serve as a starting point for other tracking algorithms. See lines 86-88.	4
In summary, ThirdPeak offers a GUI-based platform for visualizing and analyzing 2D and 3D tracking data, where the z position for 3D data is obtained via astigmatism.	ThirdPeak was designed to work independently of the nature of the tracking data, as it does not perform localization of particles, cells, or organisms. Instead, the software uses localization data generated by a localization algorithm of the user's choice, allowing high adaptability to many different image modalities. Please see lines 23-25 and 73-81.	5
It compiles various pre-existing tracking and diffusion analysis methods and presents them in a convenient drop-down menu format. Moreover, users have the option to conduct analysis on individual tracks or regions of interest directly through the GUI interface. Unfortunately, however, there is some lack of clarity with respect to the scope, as well as the limitations, of this platform. This platform also does not generate any novel statistics or advance on any pre-existing track analysis methods.	With this software, we aim to facilitate 3D track analysis across a wide range of scales, making it accessible to scientists working in various (biological) fields. The software is designed to be user-friendly and adaptable to different data ranges. Given the typically rather short lifespan of some software, we wanted to create a solution that operates independently of specific localization and tracking processes. This flexibility allows for easy adaptation to new emerging tools in the microscopy and imaging field, as the track format (X/Y/Z/T) generally remains consistent. Our advancement lies also in providing complex analysis capabilities for 3D data to untrained users and in providing an	6

	accessible toolbox to support future analysis pipelines. We clarify this in the manuscript further in lines 71-73.	
Overall impression of the work: Most of the focus of ThirdPeak is on two functions: Validation and Analysis. Its main asset is its ability to handle 3D analysis, which is not common in similar tools. Yet, it must be noted that the 3D data handled by ThirdPeak are the type where the z-position is encoded by the PSF shape. While there are many experimental setups now that treat the z-position in this way, and so ThirdPeak would have good range of applicability, this point is not stated explicitly in the manuscript.	As already stated in the comment above, this a misunderstanding. ThirdPeak is designed to work independently of the underlying microscopy data. We have made this more clear in the text in lines 71-73 and 79-81.	7
This must be stated explicitly, and the authors must clarify whether ThirdPeak can or cannot help with the visualization of 3D stacks, as some other software packages do (e.g., u-track3D by Roudot et al. [cited in the manuscript]).	While u-track 3D is the better overall solution for 3D stack data, some users might require more flexibility in their data analysis, as 3D stacks are not always generated. In such cases, ThirdPeak offers a more adaptable solution, as it can be easily integrated into the downstream analysis of most workflows and its data used further for analysis, as nearly all data at any given time can be conveniently exported as a CSV file. Please see lines 88-89; 135-137 and 143,	8
The Validation functionality allows users to visualize localizations and tracks obtained from other software and check the accuracy of these methods. This feature is not exclusive to ThirdPeak, and is available in various software packages such as SMAP (1), U-track (2D/3D)(2, 3), Spot-On(4) and many others (5, 6).	This is correct and referenced in line 96.	9
In addition, it is worth noting that the assessments via ThirdPeak are predominantly visual, and hence subjective. In contrast, for example, u-track 3D(3) offers the option to calculate a trackability score for assessing tracking performance, particularly beneficial for 3D tracking evaluations. The authors must comment on this. Where ThirdPeak stands out is in its ability to work with localizations and tracking data from external sources.	ThirdPeak offers the capability to seamlessly handle tracking data from various external sources, whereas other software often performs all necessary tasks internally, creating a somewhat closed-off ecosystem. At its current state, SWIFT does not provide an explicit trackability score, which would indeed be valuable information. Users are free to choose the tracking software they trust the most. As	10

	explained before, the development of localization and tracking algorithms are not in the scope of ThirdPeak. Please see lines 97-100. However, visual inspection of track data still remains a crucial step. Through visual inspection, we can typically interpret whether the links made by the tracking algorithm are plausible or if the parameters used for tracking result in improbable connections.	
ThirdPeak's usefulness lies in its GUI-based analysis platform, simplifying the generation of track statistics and heatmaps, particularly for 3D data. ThirdPeak offers users an opportunity to generate track statistics based on the selected ROI or single tracks. This ability to select single tracks or regions of interest through a GUI for post processing is an exciting advancement.	Thank you	11
A broader concern regarding this work lies with the description of the different features offered by the platform. The text and examples provided lack clarity regarding the extent and limitations of the platform's capabilities. A major concern is that the tracking software used by the authors, SWIFT, is not yet available and is indicated as being in preparation. This makes it difficult to address these concerns effectively. Additional clarification on these specific issues is detailed in the comments section below.	SWIFT is not mandatory but an example of the tracking software used in this project. Please see lines 85-88.	12
1. The authors need to define the minimum amount of information needed by the platform to generate track statistics and create figures. It is not clear from the current text to what extent ThirdPeak depends on external software (like SWIFT) for some basic calculations and plotting their figures. This is especially important for the custom input option (as seen in GUI). Are some inputs optional? Here are some examples highlighting the source of this confusion.	ThirdPeak accepts a plain track format (X/Y/Z/T) but can also handle an extended format generated by SWIFT. In its revised, more recent version, ThirdPeak integrates with DeepSPT by the Hatzakis lab, utilizing neural networks to generate track statistics and classify tracks based on their diffusion behavior. If only the plain track format is selected, ThirdPeak internally calculates additional data required to populate the figures. Essentially, any data not specified with "SWIFT" or "DeepSPT" is calculated internally within the software. Please see lines 73-76.	13

	We also added more description to the manual.	
a. Jump distance: Line 61 of the main text states that the platform can calculate the jump distance. Indeed, Jump Distance calculation is included under the “Functions in the software” section. However, GUI seems to also have a “jump distance” input.	This is correct. Jump distances can be calculated internally or loaded from the external tracking results if available. Please see lines 111-113.	14
b. Diffusion coefficient: Is it primarily obtained through their own calculations (lines 359-379) or by SWIFT (line 227)?	Several options are available. Either the calculation inside ThirdPeak using a classical MSD approach, a fit to the 1D jump distance distribution, a fit to the (cumulative) jump distance distribution or externally via Swift/DeepSPT.	15
c. Diffusion classification: It appears that the appearance of distinct diffusing populations is primarily performed by the SWIFT (line 216) algorithm, but their own calculations towards classification based on jump distance (line 62-62, 93-94) are also mentioned.	This is true. Swift mainly performs the classification. In case of a plain tracking data set, jump distance population classification can be used. Alternatively, the DeepSPT functionality can be used. Please see line 110-113.	16
2. Further, related to the scope of the platform, does the 3D visualization and analysis feature work with 3D stacks generated by taking z sections?	As ThirdPeak operates independently of the localization algorithm used, it can effectively handle track data generated from 3D stacks. While the visualization function for the image data of 3D timeseries was previously unavailable, we have now implemented it in the revised, updated version, anticipating it will be beneficial for the user. Please see line 96-97.	17
3. Authors need to better describe SWIFT, as it is only “manuscript under preparation”, according to the authors’ citations.	We have supplemented the manuscript with additional explanations regarding the software in the methods. However, since we are not the original authors, our ability to provide more qualified information on this matter is limited. Please see lines 309 + (79-86)	18
4. While visual comparison between dataset subsets is helpful, it is important to also have access to the actual data used in the visualization, in order to run e.g. statistical tests. This is particularly relevant in the context of running analysis on subsets of the data (ROIs or single tracks). It is not clear at present if THirdPeak currently offers access to the dataset subsets used for visualization.	ThirdPeak enables users to save the generated figures in their preferred folder. Additionally, within the same dialogue, users can extract bin edges and bin sizes from histograms, along with the underlying dataset used for histogram generation. Alternatively, all imported and calculated data available within the	19

	software can be exported through the 'Data export' tab, generating accessible CSV files for further processing if needed. We added this to the manuscript in the following section: Lines 135-138. And in the manual.	
5. The understanding of the scope of this work will be greatly improved by a more in-depth explanation of how ROI track statistics are calculated. E.g., in case the ROI happens to include only a snippet of the track/tracks, does the track statistic reflect that of the entire track or only the snippet?	In the current state, if a single point of the track is included in the region of interest (ROI), the entire track will be considered as part of it. We have introduced an additional switch in the GUI to enable both explicit and implicit selection of data points. Furthermore, we have provided explanations in the GUI, as well as in the manual and the manuscript, within the following section: Lines 106-108.	20
6. The diffusion coefficient of the simulations appear to be larger than that of experimental data (supplementary table 1). But authors claim that the difference is not large (line 227 and 83-84 respectively). How did the authors come to this conclusion?	We indeed have been unclear in this regard. We were focusing more on the high, first diffusion coefficient rather than the effective diffusion coefficient. As for the experimental data, the slow confined particles can lead to an overall slow diffusion coefficient, the fast fraction has a similar fast speed. In the simulations, the slow fraction is smaller, having less influence onto the effective diffusion coefficient. See lines 201-205.	21
7. Figures: a. Figure 1: Clearly marking out the steps that happen outside of this platform or require input from external sources will aid in understanding the platform better.	We marked the external sources and added a legend. See figure 1 in line 152.	22
Fig1C gives the impression that after looking at the tracking/localization overlays one can refine the tracking or localizations within this platform. But this is not the case, after trying out the software. Thus, in its current form, the description of ThirdPeak might be misleading.	ThirdPeak automates tracking with SWIFT by computing track statistics such as mean step length, bleaching probability, and diffraction limit. These newly calculated values are then utilized for follow-up tracking approach within SWIFT, representing a refinement process. We changed the text in the following section: Lines 167-170.	23
b. Also in Figure 1: What do the dashed gray lines in this figure mean? Finally, it is not clear why	We improved the figure to be more clear.	24

“localizations” and “tracks” branch out separately out of the “Data Import” box. Isn’t localization the first step of tracking? In addition, the “Track refinement” step after “correct drift” seems out of place, as there was no tracking step before it, in order to get tracks to refine.		
c. Supplementary figure 3: The figure showing spline fit of calibration bead stack is confusing, why is there an increase in intensity as the no of frames increases? What are frames here, are they the 10nm z steps described in materials and methods? An appropriate legend and further explanation would be helpful.	We improved the figure and added more explanation to it to increase clarity.	25
8. Simulation: a. Please provide some details about the simulation parameters used for this work.	We added the simulation parameters in a supplementary table in the methods section. See line 309 + (24-25).	26
b. In their reporting of the simulated data results, especially on lines 90-92, the authors present the results of analyzing the simulated data as if they were a “discovery”. But these are simulated data that the authors generated, and they must know the answer. The authors should comment on how well their analysis method was able to reproduce the diffusion coefficients used to generated the simulated populations.	We added a section to the manuscript explaining the data and statistics to the supplements. See lines 201-205.	27
c. In lines 86-88, the authors speculate that bleaching or confinement in the experimental data could underlie the observed differences between experimenta and simulated data. But, could tracking errors have also contributed to the differences, as experimental data is noisy. Was any noise added to the simulation to recreate noise seen in experimental data sets?	No noise was added to the simulated dataset to have pristine conditions as a reference. This additional information was added to the simulation parameter description. See lines 178-179.	28
d. The area of very low velocity for the simulated data needs further investigation and explanation. Again, these are simulated data where the authors have full control over the simulation.	Due to the nature of the simulation, slow moving Brownian particles were included. As they move slower and randomly, they are less likely to move out of a specific region. This leads to a significantly increase of slow diffusion coefficient values contributing to the local diffusion coefficient in the plot. We show this effect in an additional supplementary figure. See lines 211-213.	29
1. Figures: a. Please provide panel identifiers, they are missing from most of the figures and please provide appropriate (more detailed) legends.	We have added identifiers and more detailed legends.	30

b. Normalizing the histograms and providing the same range for the color bar for the heatmaps would be very helpful in figures where multiple datasets are being shown and compared.	We have normalized the histograms. Note that this can only be done between experiments by saving .fig files in matlab and adjusting the color scaling afterwards.	31
c. Figure 2: Diffusion coefficient units are missing. Also, in this figure, the bottom-left subpanel of each panel is called “velocity map”, while what appears to be the same figure is called “drift map” in Fig. S4.	The unit for the diffusion coefficient has been added in the revised, more recent version. In the supplementary figure however, the drift is shown as a quiver plot, as the drift values are vectors containing a direction, while the velocity is a scalar value without direction.	32
2. Please note that the following paper is now published- Roudot, P. et al. u-track 3D: measuring and interrogating dense particle dynamics in three dimensions. bioRxiv 2020.11.30.404814 (2022) doi:10.1101/2020.11.30.404814. Please cite the published version. Also, it will be helpful for the text clarity if this software is referred to as u-track3D throughout the paper, to distinguish it from the original u-track software, which did not provide visualization for 3D images. Please also discuss in more detail the differences between ThirdPeak and u-track3D.	Thank you for spotting this mistake. We changed the reference to the updated version. Since we are not utilizing 3D stack data, u-track 3Ds applicability for our purposes was limited. It's important to note that microscopy stack data is not always available or generated; for instance, a two-camera setup can also provide 3D data, as seen in zebrafish studies. Additionally, holographic microscopy data presents another scenario where no stack data is initially generated, yet it encodes 3D data. While a trackability score is advantageous, biological situations can sometimes be, confined, and limited in continuous tracks, which can decrease trackability. Binning offers a solution to reduce noise by averaging over many data points.	33
3. The ability to use this platform to compare between multiple datasets in a single figure or plot where applicable would be desirable, as often people are trying to compare these statistics between different conditions. Such a feature would increase the adoption and reach of ThirdPeak. It is not clear from Supplementary figure 5, if the option of being able to choose multiple files for analysis is for batch processing or can one compare between multiple data sets.	It is indeed possible to compare datasets within ThirdPeak. As correctly stated, users can select dataset combinations in the data analysis tab and generate figures accordingly. Additionally, the data combination leading to the figure can be saved as a CSV file, facilitating dataset comparison. Given the significant variability in data modalities across datasets and analyzed properties, creating a high-quality plotting environment to cover all cases would require a substantial amount of effort. In	34

	contrast, already available selective data export and import into another software appears to be a more feasible and versatile approach for the user.	
4. Please provide some videos (raw data, detections, tracking) of the experiments performed on T.Brucei. It will help readers assess the quality of data, get a sense of how dense the localizations are, especially because the dye is labelling all surface glycoproteins.	Added a supplementary figure with time series of the original data, the detections and the tracking	35
5. What were the localization and tracking parameters for the simulation and experimental data sets? Were they the same?	They were the same for most of the parameters. During localization with SMAP, we used an acquired z-stack of fluorescent beads as a calibration measurement. For the simulated data, we also simulated an artificial bead stack using the same laser and PSF modalities as we could not reproduce a perfect copy of our microscope PSF in the software. For tracking, the same initial parameters were chosen. As ThirdPeak automatically determines better fitting parameters after the first tracking iteration, the final tracking parameters will differ to be most appropriate for the respective dataset.	36
6. If the diffusion coefficients are calculated only in 2D (line 228), then where is the 3D diffusion coefficient calculation used (line 378-379)?	The diffusion coefficient is calculated in 2D and 3D if 3D data is available. If only 2D data is available, only 2D data can be used. We provide several options to determine the diffusion coefficient. Either by external values from the tracking algorithm or DeepSPT. Inside ThirdPeak we can perform a 1D step analysis which is applicable for a single, normal distribution behavior. The classical MSD analysis can be performed in 2D and 3D as well as the mean jump distance distribution fit.	37
7. The title of the section: Confinement Ratio vs mean jump distance does not match with the description (line 385-389). There is also another reference to the confinement ratio in line 356.	We have changed the heading to “Basic confinement ratio” which compares the track lengths, while the “diffusional confinement ratio” uses a diffusion-based approach to determine confinement, which, however, is computationally expensive	38
8. On trying the software, the manual was found to contain very little information regarding the use of the canned data. The manual can be made more user	We have extended the manual on this section and provide a “Quick-start guide”. We now also provide the whole data used on zenodo for further exploration	39

friendly by giving examples for every step using the canned data.		
Reviewer #2 (Remarks to the Author):		
The paper broadly compares the capabilities of ThirdPeak against the competition, however it is not clear how each part of the pipeline differs from the other tools that are cited.	We aim to provide a comprehensive overview of available tools for new users who are looking to establish their own pipeline, as our software acts as a toolbox to build upon them. We do not present a software that wants to compete against existing tracking algorithms. Instead, ThirdPeak, builds on them by offering the possibility of integrating existing and future algorithms. Thus, ThirdPeak is an integrative toolbox that allows the users (even unexperienced ones) to analyse their 2D and 3D-tracking data with highest accuracy. At the moment we do not see a comparably versatile and adjustable solution.	40
The feature reporting is also sometimes misleading, since both Kuhn and Roudot's paper propose automated approach for validation while Muller's is manual.	We are not developing a localization or tracking algorithm; instead, we utilize SMAP/DECODE and SWIFT. Visual inspection of tracks is a universal approach applicable to all tracks across dimensions and available for all image modalities. Specific tracking algorithms offer better metrics, such as the trackability score from u-track 3D, but it was never in the scope of ThirdPeak to establish a new localization or tracking algorithm by itself. We clarify this in the manuscript in lines 96-100.	41
As such, it appears that the software does not provide conceptual advances but a new package to load, track, visualize and analyze the data. This is in itself can be very valuable, especially considering the dearth of trajectory analysis tools.	To our knowledge, analyzing and comparing noisy, biological 3D track data of various scales has not been addressed by existing packages. Many of these packages typically require imaging data and perform all processes internally, which simplifies usage but limits flexibility. By decoupling the processes, ThirdPeak offers greater adaptability to diverse situations and is not solely restricted to fluorescent data. Please see lines 245-250.	42
However, it is impossible to evaluate the value of the new software without actually trying it. To that end,	A small subset of data is already available on the GitHub page of the software	43

I would need the authors to provide a minimal example of input to test the software.	("TestDataAndSettings"). Additionally, all additional data is now accessible on Zenodo. However, we also encourage users to utilize their own data to test the adaptability of our software, as we cannot account for all the diverse microscopy modalities.	
I guess I could try trackIt with my own data to test it, but I respectfully believe that this is out of the scope of the reviewer duty.	We absolutely understand that this is out of scope for the reviewer.	44
The motivation of the work is in the introduction is unconvincing at time: - Why taking about PSF engineering in particular ?	We use it for our study just as an example, as it offers one method of generating 3D localization data. However, it is important to note that different experiments require different microscopic setups, thereby also generating different imaging data that may require alternative analysis pipelines. We extended the introduction to allow for a smoother transition into the main text. Please see lines 17-25.	45
- TrackMate and others do not need SMAP or Picasso to produce tracks, they can do so readily on volumetric sequence.	The scope of the software extends beyond volumetric sequences, as astigmatic data, which is not volumetric by default, is also accommodated. SMAP and Picasso are tailored for single-molecule imaging conditions. Additionally, tracking organisms with two perpendicular cameras does not generate classical volumetric image data but still contains information of motion in 3D space. Moreover, digital holographic microscopy data is not directly interpretable as volumetric data and requires preprocessing and further segmentation, as brightfield-like images are usually generated. We added a paragraph to address volumetric imaging to summary in lines 245-250.	46

REVIEWERS' COMMENTS:

Reviewer #1 (Remarks to the Author):

I thank the authors for the substantial effort they put into revising the manuscript and clarifying what their software does (and does not do). They have addressed my previous comments, and I have no further critiques.

Thank you again for your valuable input.

Reviewer #2 (Remarks to the Author):

I appreciate for the authors revision effort. One remaining important for me is to make sure the software is usable. I currently get stuck at the step highlighted in the file attached. I am happy to try again or discuss debugging.

We are grateful for the thorough review. The primary appeal of this software is its usability, so it's essential that issues are promptly addressed. However, without an associated error message, debugging based solely on the screenshot is challenging.

ThirdPeak has been used in our department for the analysis of imaging data and was performing well during localization filtering. The software will open the localization data if the manual filter checkbox is selected during filtering. One might use the space bar to activate the manual filtering, position points by clicking with the left mouse, and finish the process by double clicking or by pressing enter. Since multiple cells could be present in the image, the same image as before will open, allowing to efficiently segment the data of multiple cells. If no further data should be masked, pressing enter twice will stop the masking. After that, the next localization file will be opened and so on, until all files are processed. After that, ThirdPeak will calculate some parameters necessary for tracking.

As no direct problems could be seen in the code, a possible explanation could be a misunderstanding when using the software. To clarify the occurrence of repeating images, which could be interpreted as an error, we adjusted the QuickStart manual and the full manual to prevent misunderstandings. We also conducted some small bugfixes on the way and compiled a new version, which can be found on Github.

If problems arise during tracking, swift might not be installed on the system, so it cannot be used by ThirdPeak. This however should only occur after the masking step is performed.

If the problem persists, efficient bug fixing is probably best realized using Github as it allows for more dynamic responses.